# Undamming the Douro River Catchment: A Stepwise Approach for Prioritizing Dam Removal

**Rui M.V. Cortes** [1], **Andrés Peredo** [2], **Daniela P.S. Terêncio** [1],
**Luís Filipe Sanches Fernandes** [1], **João Paulo Moura** [1], **Joaquim J.B. Jesus** [1],
**Marco P.M. Magalhães** [1], **Pedro J.S. Ferreira** [1] and **Fernando A.L. Pacheco** [3,*]

1   Centro de Investigação e Tecnologias Agroambientais e Biológicas, Universidade de Trás-os-Montes e Alto
    Douro, Ap 1013, 5001–801 Vila Real, Portugal; rcortes@utad.pt (R.M.V.C.); dterencio@utad.pt (D.P.S.T.);
    lfilipe@utad.pt (L.F.S.F.); jpmoura@utad.pt (J.P.M.); jjesus@utad.pt (J.J.B.J.); mpmmaga@utad.pt (M.P.M.M.);
    pedrof@utad.pt (P.J.S.F.)
2   Instituto Superior de Agronomia, Universidade de Lisboa, Tapada da Ajuda, 1349–017 Lisboa, Portugal;
    andresperedoarce@gmail.com
3   Centro de Química de Vila Real, Universidade de Trás-os-Montes e Alto Douro, Ap 1013,
    5001–801 Vila Real, Portugal
*   Correspondence: fpacheco@utad.pt

**Abstract:** Dams provide water supply, flood protection, and hydropower generation benefits, but also harm native species by altering the natural flow regime, and degrading the aquatic and riparian habitats. In the present study, which comprised the Douro River basin located in the North of Portugal, the cost-benefit assessment of dams was based upon a balance between the touristic benefits of a dammed Douro, and the ecological benefits of less fragmented Douro sub-catchments. Focused on four sub-catchments (Sabor, Tâmega, Côa and Corgo), a probabilistic stream connectivity model was developed and implemented to recommend priorities for dam removal, where this action could significantly improve the movement of potadromous fish species along the local streams. The proposed model accounts for fish movement across the dam or weir (permeability), which is a novel issue in connectivity models. However, before any final recommendation on the fate of a dam or weir, the connectivity results will be balanced with other important socio-economic interests. While implementing the connectivity model, an inventory of barriers (dams and weirs) was accomplished through an observation of satellite images. Besides identification and location of any obstacles, the inventory comprised the compilation of data on surrounding land use, reservoir water use, characteristics of the riparian gallery, and permeability conditions for fish, among others. All this information was stored in a geospatial dataset that also included geographical information on the sub-catchment drainage network. The linear (drainage network) and point (barriers) source data were processed in a computer program that provided or returned numbers for inter-barrier stream lengths (habitat), and the barrier permeability. These numbers were finally used in the same computer program to calculate a habitat connector index, and a link improvement index, used to prioritize dam removal based upon structural connectivity criteria. The results showed that habitat patch connectivity in the Sabor, Tâmega and Côa sub-catchments is not dramatically affected by the installed obstacles, because most link improvement values were generally low. For the opposite reason, in the Corgo sub-catchment, obstacles may constitute a relatively higher limitation to connectivity, and in this case the removal of eight obstacles could significantly improve this connectivity. Using the probabilistic model of structural connectivity, it was possible to elaborate a preliminary selection of dams/weirs that critically limit stream connectivity, and that will be the focus of field hydraulic characterization to precisely determine fish movement along the associated river stretches. Future work will also include the implementation of a multi-criteria decision support system for dam removal or mitigation of the critical structures, as well to define exclusion areas for additional obstacles.

**Keywords:** dam removal; habitat patch connectivity; fish migration; probabilistic model; metrics

## 1. Introduction

The construction of dams affects the biodiversity of riverine ecosystems in a number of manners. The hydrologic regime of streams is modified from lotic to lentic when stream water is retained in the reservoir, while this new environment frequently promotes the anomalous spreading of exotic species more adapted to the lentic conditions, and concomitantly the fall of native species [1]. The cyclic streamflow variability of rivers is reduced by the presence of a new water body [2], which has been shown to increase water channel homogeneity and degrade aquatic fauna habitats [3]. The phytoplankton productivity and the thermal stratification of water tend to increase while dissolved oxygen decreases, resulting in a rapid increase of macronutrients in lake water [4,5] that temporarily increases the abundance of aquatic flora [6]. However, the progressive increase of nutrients in the reservoir deteriorates water quality, ultimately leading to eutrophication with negative consequences for the aquatic fauna [7–10]. Water quality deterioration can be particularly expressive in multiple land use or heavily urbanized catchments, watersheds affected by recurrent wildfires, or basins affected by inadequate land use changes [11–16], aggravating the impacts on aquatic fauna.

Concurrently with the impact on stream flow regime and freshwater quality, the damming of rivers hinders the free circulation of migratory or resident species, causing habitat fragmentation. The interruption of connectivity reduces the abundance of spawning sites compromising the survival capacity of juveniles [17]. This ability to survive can be further reduced downstream from the dam by hypolimnetic discharges and concomitant fluctuations in temperature and stream water composition [18,19]. The longitudinal dimension of river connectivity has been significantly disrupted by barriers in all Mediterranean areas to compensate for water demand in the long periods of water scarcity. These changes are also widespread on the Portuguese side of Douro River drainage network [20–22], which will be used as our study site. In this case, side by side with agriculture intensification, there was a pressure to increase hydropower production, especially in the last decades. In fact, this pressure continues because new complex systems are still under construction (e.g., in the Tâmega River). Therefore, it is difficult to find a stream segment in the Douro that is not virtually obstructed by at least one dam or weir.

The scenarios of generalized habitat fragmentation urge mitigation. A review on the ecological restoration of fluvial rivers has recently been published [23], which comprises techniques for rivers affected by engineering control. In general, the ecological restoration of rivers affected by the construction of hydraulic facilities mainly includes two aspects: Restoration of the river's geomorphological features, and the natural hydrological regime. The restoration zones of river geomorphological features can be divided into two parts: Riparian restoration and river channel restoration [24,25]. The natural hydrological regime can be restored through reconnecting abandoned channels to mainstreams, restoring the links between surface and groundwater flow to enhance vertical connectivity and communities associated with the hyporheic zone, and decommissioning of unsafe or obsolete dams, which can take the forms of full removal, partial removal of key components, or abandonment [26]. New concepts on the spatial planning of sustainable water reservoirs [27,28] can also be used to reconcile water resources management and the protection of biodiversity. It is worth to note, however, that the implementation of connectivity carries out complex social and political decisions, which can also be technically challenging because of the unique dendritic spatial structure of river systems [29–31].

Investigation on dam removal has become intense, and has spanned a diversity of topics in recent years. Various studies described the restoration of fish habitats or freshwater food webs following the removal of dams [32–35]. Some other papers were more focused upon changes in the stream channel morphology which occurred after dam removal, and on potential consequences for habitats [36–43].

There were also studies interested in predictions of river aquatic productivity and stream water quality before and after dam removal [44], or highlighting promising habitat and unpromising economic use tradeoffs for water supply and hydropower generation [45]. Finally, a few works were dedicated to the inventory of obsolete dams, aiming at the assessment of their safety and structural integrity [46], to the removal of large dams [47,48], or to the pros and cons of dam removal as regards climate change, landscape aesthetics or social impact issues [49–51].

The abundant literature on dam removal impacts allowed a rapidly expanding body of literature, describing the application of spatial planning to inform river authorities. A compilation on connectivity prioritization decisions is reported in the review work of McKay et al. [52].

Despite the large number of papers published on the impacts of dam removal, the implementation of methods to prioritize obsolete dam or weir removal is, however, poorly developed. In fact, only a few studies directly addressed this topic in the recent past [53,54]. The model by Kuby and his co-workers [53], published in 2005, was focused on salmon migration in the Pacific Northwest region of the United States, and aimed to optimize ecological and economic loss objectives. The modeling of an ecological objective had the purpose of maximizing the river and stream drainage area that has migratory access to the ocean. The economic loss objective was assumed to be purely illustrative in the model, and aimed to minimize two of the many economic services that are lost when a dam is removed: Hydroelectric generating capacity and storage capacity. Among the specificities of Kuby's model, one is noticeable: The function used to maximize the ecological function ensures that the benefits of removing a dam are not counted, unless salmon would encounter no other dams between the ocean and the removed dam.

The more recent model of Hoenke, Kumar and Batt, published in 2014 [54], comprised a GIS tool for prioritizing dams for removal based on ecological criteria, social and ecological criteria taken altogether, and habitability of anadromous fish criteria. The results differed among the three model runs, but some dams ranked high in all runs, and the reason as to why were considered a top priority for removal. The tool includes a criterion for improving habitat connectivity based on three connectivity indicators: (a) Functional upstream mileage, which is the number of miles upstream of a dam to the next barrier; (b) total upstream and downstream functional connectivity, which is the combined number of functional miles upstream and downstream of the dam; and (c) downstream barriers, which is the number of barriers downstream of a dam. A higher indicator rank is assigned for higher upstream mileage and total connectivity. For anadromous fish prioritizations, a higher rank is assigned for a lower number of downstream barriers.

The aforementioned and other similar models lack considerations about the permeability conditions across dam structures when the criterion for any improvement of habitat connectivity is calculated. This would include, for example, information on the presence (or not) of passages for fish and their conditions (full, partial or non permeability). The purpose of this study was therefore to improve the assessment of habitat connectivity through the definition of a probabilistic function, where permeability is taken into account. The model is then tested in four sub-basins of the Douro River Basin located in Portugal. Since the work of this research team aims to cover all of the Douro River Basin, and accounts for a multitude of other procedures not addressed in this manuscript, this study is primarily interested on exploring the probability function capabilities, while the application results may be viewed as preliminary.

## 2. Materials and Methods

### 2.1. Study Area

The Douro River is an Iberian water course (Figure 1a). The catchment headwaters are located in Spain, in the Urbion Mountains (Cordillera Ibérica), and rise up to 1260 m (ca. 4134 feet). The main water course is 927 km (576 miles) long, and debouches into the Atlantic coast in Portugal, close to Porto town.

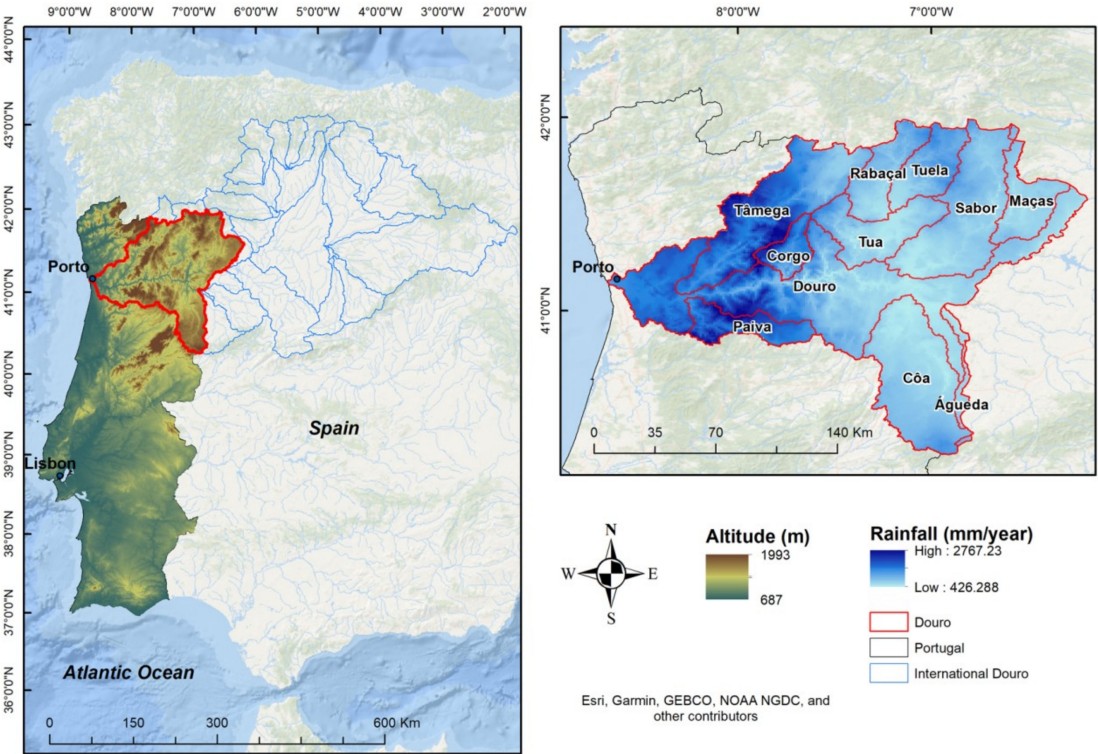

**Figure 1.** (**a**) Location map illustrating the Douro River Basin and evidencing the Portuguese side over the topographic map. (**b**) Portuguese side of the Douro River Basin evidencing the most important tributary catchments and spatial distribution of annual rainfall.

The Portuguese Douro River Basin covers an area of approximately 19,000 km$^2$ (ca. 7336 square miles). The population living in this area reaches two million, distributed within 74 municipalities. The catchment comprises nine sub-basins: Águeda, Côa, the coastal area between the Douro and Vouga, Douro, Paiva, Rabaçal/Tuela, Sabor, Tâmega and Tua. The Águeda, Douro, Rabaçal/Tuela, Sabor and Tâmega sub-basins are transboundary river basins, while the Côa sub-basin corresponds to a frontier river basin.

According to the Koppen-Geiger classification, climate in the Douro watershed is temperate, with well-defined seasons, and average annual temperature approaching 13 °C. Average rainfall in the catchment comes near 1000 mm (over 39″), varying between 541 mm (21.3″) and 1773 mm (70″), as a function of topography and the distance to the Atlantic Ocean (Figure 1b). The average relative humidity in the region is 71%, ranging from 58% in July to 82% in January (https://www.ipma.pt).

A large number of surface water bodies are distributed within the Portuguese Douro watershed, namely 361 rivers, 17 reservoirs, three transitional waters and two coastal waters. The available surface water resources approach 8023 hm$^3$/year. A portion of this water (1594 hm$^3$) is stored in 67 large dams and a huge number of small dams or weirs. In most large dams the barrier effect is problematic, because the large heights hinder the placement of effective devices for the transposition of aquatic fauna, namely fish. The smaller dams or weirs cause a barrier effect, but this is often mitigated by the existence of those devices (https://www.apambiente.pt).

*2.2. Methodology*

The method we used to prioritize dam removal follows the steps of the McKay et al. model [52], which generally comprise: 1) Set the scope, 2) develop a geospatial database, 3) predict connectivity for the watershed, 4) compute costs and benefits of alternative scenarios, 5) summarize information for decision making and take action, and 6) do not forget post-project actions. The full inventory of these steps is portrayed in Figure 2 in the form of a general workflow. This figure highlights steps

1 to 3 covered in this study. As mentioned above, the current objectives are focused on exploring a connectivity model to set up a dam removal priority. Although the general steps are inspired by this work of McKay and his co-workers, the specific steps were adjusted to the study area (steps 1 and 2) or introduced and implemented in this study (Step 3).

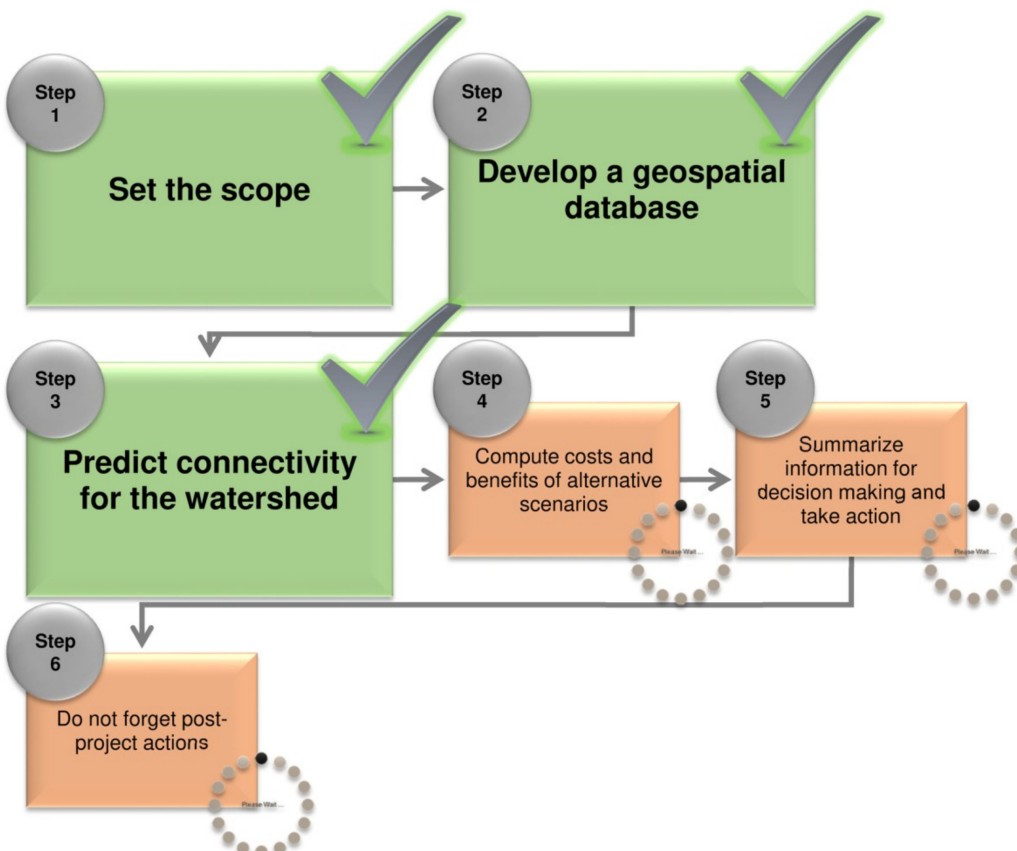

**Figure 2.** General workflow for prioritizing dam removal. Steps 1–3 are addressed in this study, while steps 4–6 correspond to ongoing research.

### 2.2.1. Identify the Scope of the Analysis

Step 1 is focused on the characterization of the Douro Basin's stream network (Portuguese side). The selection of this Portuguese Douro catchment allows extending a similar work already accomplished on the Spanish side (http://www.chduero.es/acciona5/metodologia/ic.pdf). Besides, it helps in responding to the increasing concerns about biodiversity losses caused by high fragmentation, and new projects under construction in this large watershed [55–57], and it represents an opportunity to update the information on the location of stream obstacles (dams and weirs) as well as their attributes (relative impacts, use, conservation, licensing or ownership), which is lacking or incomplete.

The characterization step spanned the compilation of digital data on catchment limits (Douro Basin and main tributaries) as well as on the corresponding drainage network; the location of dams and other obstacles as well as of associated reservoirs and their extension; the preliminary assessment of riparian galleries and urban areas in the neighborhood of reservoirs; and finally the land use along the main tributaries. The main water course (the River Douro) was excluded from the analysis. This decision was considered inevitable, but did impose some restrictions to the connectivity analysis. The issues and implications involved in this decision were: 1) The large Crestuma-Lever dam, installed in the Douro River close to the estuary, virtually impedes the migration of diadromous species such as Atlantic salmon or European brook and river lampreys. The fish passages are ineffective,

and there is a sharp salinity contrast between the fresh river water upstream of the dam, and the brackish water downstream. However, it is not considered possible to remove the dam because of its importance for hydropower generation. On the other hand, the use of the Douro River is crucial for local socio-economic reasons, where tourism has become the main driving force, and the dams contribute to that economy, besides other multiple purposes.

It is therefore inevitable to be required to keep the large infrastructures of the Douro on site. Facing the aforementioned options, the current assessment on connectivity will not cover the benefits of dam removal for the Diadromous species. The rationale differs for Potamodrous species such as brown trout, because the other Douro River dams are equipped with fish passages, which regardless of poor maintenance, allow the movement of fish. In this case, the connectivity along the main water course is less affected.

### 2.2.2. Develop a Geospatial Database

The geospatial database (Step 2) was developed during 2018 to store all information related to the project. The operational steps are illustrated in Figure 3. Firstly, the hydrographic and drainage information was extracted from the European EU-Hydro basis, which was developed under the Copernicus program https://land.copernicus.eu/pan-european/satellite-derived-products/eu-hydro/eu-hydro-public-beta/view. This information is available in vector format (geodatabase), and contains high resolution drainage network elements such as basins, catchments, drainage lines and nodes; it also includes dams, coastlines and land polygons. Secondly, satellite images from Google™ and Bing™ were visually interpreted to obtain data on the location of obstacles, type of barrier (dam or weir) and its physical characterization (e.g., material used in the construction, presence of fish passage or water mill), as well as any surrounding settings (e.g., type of riparian gallery 100 m upstream and 100 m downstream from the site). Thirdly, complementary information was interpreted from military maps (e.g., quality of road access; https://www.igeoe.pt), or compiled from institutional sources. In the latter case the complementary information included land use in the 1 km buffer obtained from COS2015 cartography (http://www.dgterritorio.pt/), or data about fish passage characteristics in large dams obtained from the National Institute for the Conservation of Nature and Forests (https://www.icnf.pt/). The full check-list of parameters included in the geospatial database is provided as Supplementary Materials.

For the present study, an important parameter to consider was the permeability of obstacles, classified in the 0–1 range. The classification of permeability was performed according to the criteria described in Table 1. The maximal value (one) represents full permeability, and occurs when the fish can move freely along the water course (absence of barrier). Permeability decreases whenever the fish encounters a barrier. If the barrier was equipped with a fish way, the decrease was considered small, and therefore the permeability value was reduced to 0.9. If the observation of Google™ or Bing™ images identified a barrier but not an upstream reservoir, it was assumed that the obstacle allows stream flow, and therefore the permeability value was reduced to 0.7. When the available information confirms the existence of a reservoir created by the barrier, but the reservoir could not be observed in the satellite images the permeability value was reduced to 0.5. Finally, for barriers with reservoirs clearly detected in the satellite images the permeability value was reduced to 0.3.

The data obtained from interpretation of images or maps, or compiled from the various complementary sources, were assembled in two vector databases using QGIS (Figure 3). Subsequently, these datasets were processed in a computer package for the calculation of connectivity indicators (Step 3), as described in the next section. The results were joined to the drainage network and obstacle geodatabases for presentation of their spatial distributions. The thematic maps were produced in the ArcMap software of ESRI (https://www.esri.com), commonly used in many environmental contexts [56–67].

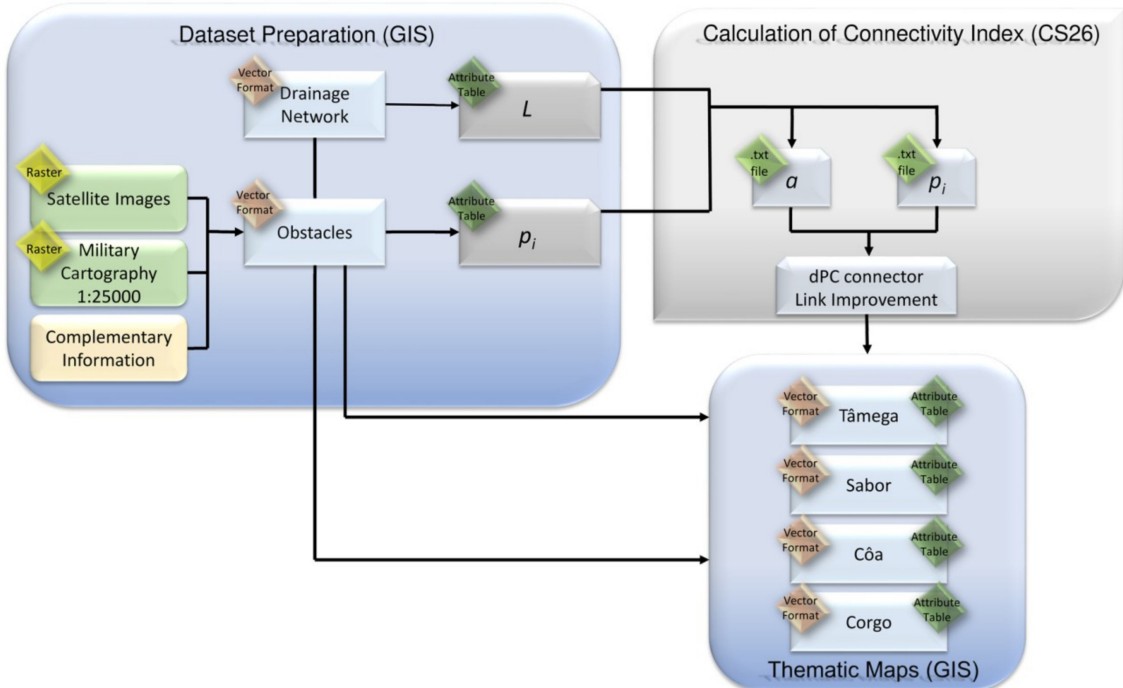

**Figure 3.** Workflow used to calculate connectivity indicators.

**Table 1.** Permeability conditions for fish in the obstacles (dams or weirs).

| Type of Structures | Permeability Value ($p$) |
| --- | --- |
| No barrier | 1 |
| Barrier with fish way | 0.9 |
| Barrier without reservoir | 0.7 |
| Barrier with reservoir (not visible in the cartography) | 0.5 |
| Barrier with reservoir (visible in the cartography) | 0.3 |

### 2.2.3. Predict Connectivity for the Watershed

The probabilistic model of structural connectivity (Step 3) resorted to the Conefor Sensinode 2.6 (CS26) computer package [68], a free software available at http://www.conefor.org. This software computes the importance of habitat patches (termed nodes) for maintaining or improving structural landscape connectivity, and is conceived as tool for decision-making support in landscape planning and habitat conservation. CS26 includes graph-based connectivity metrics hinged on the habitat availability concept, which considers the node as a space where connectivity occurs, integrating the habitat area from the nodes and the connections between different nodes (termed links). Therefore, habitat availability for fishes will be low if habitat patches are poorly connected, such as in a highly fragmented network with fragmentation caused by weirs.

The raw data for CS26 comprised the vector dataset with the drainage network characterized for stream length, and the vector dataset with the barriers classified according to permeability (Figure 3). These GIS datasets were used as input to the ArcGIS extension embedded in CS26, which produced the "nodes" and "links" text files as output. The "nodes" file contained the list of stream segments located between barriers and their lengths ($a$). The "links" file contained the barrier permeability ($p$) retrieved from the barrier vector dataset. The two files were used as input to CS26, which calculated the connectivity indexes for the segments, and the link improvement values for the barriers, according to specific equations described below. Finally, these values were joined back to the original stream and barrier vector datasets to produce the thematic maps in ArcMap (e.g., Figure 4 below).

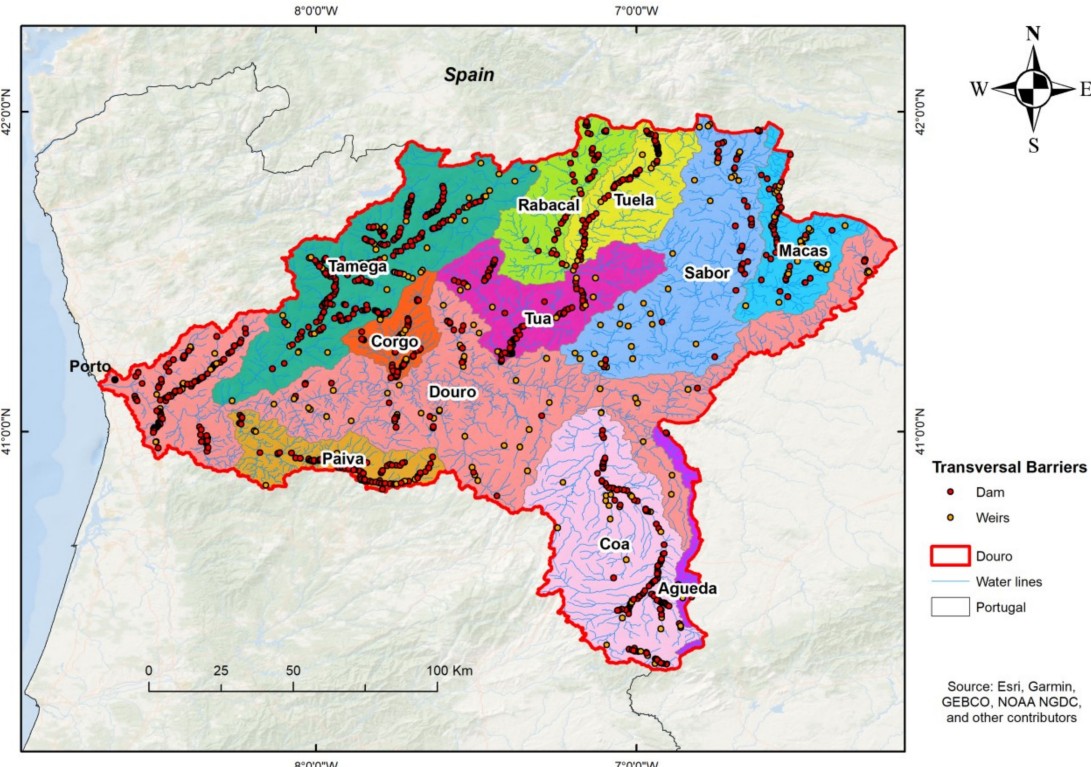

**Figure 4.** Spatial distribution of dams and weirs within the Portuguese side of the Douro River Basin.

The index used to calculate the habitat connectivity was the so-called PC index. This index is embedded in the CS26 software and comprises one among a diversity of other indices. The PC index is described as follows:

$$\text{PC} = \frac{\sum_{i=1}^{n} \times \sum_{j=1}^{n} ai \times aj \times pij}{A_L^2} \tag{1}$$

where $n$ is the total number of nodes in the landscape (the segments of drainage network between barriers), $a_i$ and $a_j$ represent the habitat in the nodes $i$ and $j$ (the length of the segments), $p_{ij}$ is the probability of connection between the patches $i$ and $j$ (the permeability value according to Table 1), and $A_L$ is the total habitat across the landscape (the total length of the river network). As mentioned above, the probability of patch connection varies between 0 and 1. The extreme values represent totally impermeable and totally permeable conditions for fish, respectively. A list of patch probabilities ($p_{ij}$) representing a diversity of permeability conditions for fish were depicted in Table 1. These values are valuable indicators of permeability, and are adequate for the evaluation of connectivity in the regional scale for planning purposes (current status). The evaluation of connectivity in the local scale and for decision making purposes requires the confirmation of $p_{ij}$ based on field work, which is in progress.

The percent variation in PC caused by the removal of node $k$ can be calculated as $\text{dPC}_k$, which may be split into three fractions: $\text{dPC}_k$ intra, $\text{dPC}_k$ flux and $\text{dPC}_k$ connector:

$$\text{dPC}_K = 100\frac{\text{PC} - \text{PC}_{\text{removeK}}}{\text{PC}} = \text{dPCintra}_K + \text{dPCflux}_K + \text{dPCconnector}_K \tag{2}$$

where dPC intra measures the habitat provided by the node itself (the length of the segment), being therefore completely independent on how this node is connected to the surrounding network. The fraction dPC flux measures how well the node is connected to other segments. Finally, the fraction dPC connector measures how much the node contributes to the network connectivity. In order to prioritize dam removal/rehabilitation, this dPC connector is an important conservation parameter: If a

given segment ranks high, it is not advisable to build a barrier there, because the system's connectivity will drop.

A final step towards the evaluation of structural connectivity was accomplished with the calculation of a parameter called link improvement. This is related to the links and not to the nodes, and shows how much the PC will improve if the probability of connection between two nodes becomes maximal (in this case, if the permeability value changes to 1). The link improvement parameter allows identifying which barriers should be prioritized for removal in order to improve the river's network connectivity. It is worth noting that the values of link improvement are negative. Therefore, the lower they are, the larger will be the connectivity increase.

## 3. Results

The distribution of dams (127) and weirs (1066) in the Portuguese Douro catchment is illustrated in Figure 4, and represent a total of 1193 obstacles. The majority of these obstacles (604) were installed in low order streams, but a large number (589) were placed in the main water courses of the Douro catchment and principal sub-catchments (e.g., Tâmega, Sabor, Côa or Corgo). The construction materials were rock fill (912), concrete (280) or earth (1). The land use surrounding the dams was agriculture (905 cases), scrubland (3), forest (1) or mixed (284). The riparian gallery were mostly sparse (1181 cases) or inexistent (1), being continuous in just 11 cases. In general, the dams or weirs were used for irrigation (1129 cases), while some dams corresponded to large (15) or small (51) hydroelectric plants.

The tested sub-catchments were four: Tâmega, Sabor, Côa and Corgo. Within these sub-basins a total of 184 obstacles were used to check the connectivity model. The density of obstacles is not uniform across the studied area. In the Corgo (3.9 obstacles/100 km$^2$) and Tâmega (3.22) basins the coverage with dams or weirs is much denser than the coverage in the Sabor (1.33) or Côa (1.19), but rainfall in the first two catchments ($\approx$1500 mm/year) is also much larger (twice) than in the other catchments ($\approx$750 mm/year). Therefore, dam or weir installation seems to be related with rainfall availability, rather than water scarcity.

The evaluation of connectivity indices dPC connector and link improvement was restricted to the four sub-basins (Figure 5). This corresponded to the calculation of connectivity indices in 1497 nodes and 184 dams or weirs. The average node lengths ranged from 2404 m (Tâmega, 468 nodes), 2908 m (Sabor, 631 nodes), 4021 m (Côa, 305 nodes) and 2590 m (Corgo, 93 nodes). These results link longer nodes to the Côa basin, and shorter ones to the Tâmega basin. The permeability indices ranged from $p = 0.68 \pm 0.11$ (Tamega, 83 barriers), $p = 0.62 \pm 0.14$ (Sabor, 48 barriers), $p = 0.65 \pm 0.10$ (Côa, 35 barriers), and $p = 0.70 \pm 0.07$ (Corgo, 18 barriers), meaning that they are relatively constant within the four basins.

The counting of habitat patches per classes of dPC connector and of obstacles per classes of link improvement are depicted in Tables 2 and 3, respectively. The barriers are located in streams characterized by diverse structural connectivity. In the Tâmega sub-basin, the majority of connected patches are linked to extremely low (423, 90.4%) or very low (38, 8.1%) dPC connector values (Table 2). The impact of removing a dam or weir in this catchment is very low in 72 cases (86.7%) and low in 9 cases (10.8%), attaining to the moderate class just in two sites (2.4%) (Table 3).

**Table 2.** Distribution of habitat patches per dPC connector classes (confirm ranges in Figure 5).

| Sub-basin | dPC Connector Class | | | | | | | | Total |
|---|---|---|---|---|---|---|---|---|---|
| | 0–5.75 | 5.75–11.5 | 11.5–17.25 | 17.25–23.0 | 23.0–28.75 | 28.75–34.5 | 34.5–40.25 | 40.25–47.0 | |
| | Extremely Low | Very Low | Low | Moderately Low | Moderately High | High | Very High | Extremely High | |
| Tâmega | 423 | 38 | 7 | 0 | 0 | 0 | 0 | 0 | 468 |
| Sabor | 554 | 27 | 16 | 6 | 5 | 8 | 8 | 7 | 631 |
| Côa | 246 | 39 | 6 | 4 | 6 | 4 | 0 | 0 | 305 |
| Corgo | 57 | 18 | 9 | 8 | 1 | 0 | 0 | 0 | 93 |

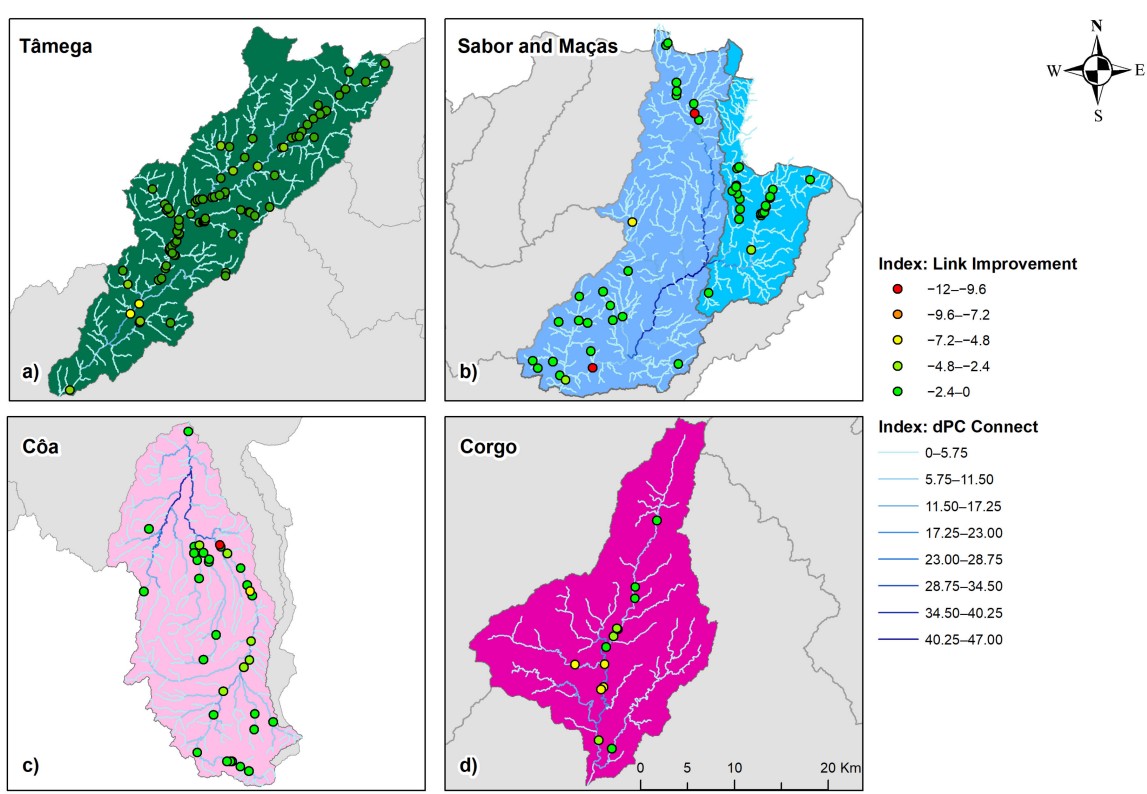

**Figure 5.** Spatial distribution of connectivity indices dPC connector and link improvement, within the tested sub-basins: (**a**) Tâmega River Basin; (**b**) Sabor River Basin; (**c**) Côa River Basin; (**d**) Corgo River Basin (on a different scale to the others).

**Table 3.** Distribution of dams and weirs per link improvement classes (confirm ranges in Figure 5).

| Sub-basin | Link Improvement Class | | | | | Total |
|---|---|---|---|---|---|---|
| | 0 to −2.4 | −2.4 to −4.8 | −4.8 to −7.2 | −7.2 to −9.6 | −9.6 to −12.0 | |
| | **Very Low** | **Low** | **Moderate** | **High** | **Very High** | |
| Tâmega | 72 | 9 | 2 | 0 | 0 | 83 |
| Sabor | 43 | 2 | 1 | 0 | 2 | 48 |
| Côa | 27 | 6 | 1 | 0 | 1 | 35 |
| Corgo | 5 | 5 | 4 | 4 | 0 | 18 |

The concomitance between the high frequency of low dPC connector values and high frequency of low link improvement is replicated for other sub-basins. In the Sabor, quite a number of patches (28, 4.4%) are moderately high to extremely high connected, and in this catchment there are two sites where dam or weir removal would increase patch connectivity substantially, and a similar scenario is also found for the Côa. The exception seems to be the Corgo sub-basin. In this catchment, connectivity is essentially extremely low to low (84 patches, 90.3%), as in the Tâmega, but the removal of eight out of 18 (44.4%) dams or weirs would result in a moderate to high increase of structural connectivity.

## 4. Discussion and Future Work

Dams provide water supply, flood protection, and hydropower generation benefits, but also harm native species by altering the natural flow regime and degrading aquatic and riparian habitat. In a study in California [45], Sarah Null and her co-authors concluded that removing all rim dams would not be beneficial to the state, but that a subset of existing dams would be promising candidates for removal from an optimized water supply and free-flowing river perspective. In the present study, the option for excluding the Douro River dams from the connectivity and removal analyses was

based on a similar rationale, namely the balance between the touristic benefits from a dammed Douro, and the ecological benefits from less fragmented Douro sub-catchments. While taking this option we were also restricting the connectivity analysis to potadromous fish species distributed within the various Douro sub-catchments, namely the Sabor, Tâmega, Côa and Corgo basins. Other studies have also directed the focus to the effects of dam removal on resident lotic fish species [69].

The general results depicted in Figure 5a–c and systematized in Table 3 show that habitat patch connectivity in the Sabor, Tâmega and Côa sub-basins is not dramatically affected by the installed obstacles, because most link improvement values fall within the lowest classes. Nevertheless, the two cases in the Sabor and one case in the Côa where the link improvement index is very high need special attention, especially because the Côa obstacle and one of the Sabor obstacles are located in the main water course and close to the catchment outlets, and therefore have substantial functional upstream connectivity. Figure 5d and Table 3 reveal that obstacles constitute a relatively higher limitation to connectivity in small rivers running directly into Douro River, such as the Corgo River. In this case, the removal of eight obstacles would considerably improve connectivity. Zooming-in on the figures also indicates that many lower ranked dams (low dPC connector and low link improvement values) in the Sabor, Tâmega and Côa sub-catchments tend to be on small tributaries that are used as agricultural ponds, and therefore have less functional upstream connectivity, while in the Corgo sub-catchment, lower as well as high ranked dams are mostly on the main water course.

The effects of connectivity improvement on fish assemblages are usually beneficial and known to develop rapidly after barrier repair, removal or retrofitting. A study in the Baraboo River [70], Wisconsin, reported that 10 out of 11 fish species, which had been entirely or mostly restricted below dams, were able to recolonize upstream sections within one year after dam removal. Two other studies conducted along rivers in Wisconsin and Michigan [71] observed significant increases in native fish abundance within 4–5 years after dam removal. A number of other studies clearly report relatively short-term increases in fish abundances, fish biodiversity metrics and assemblage structure, both upstream and downstream from the former dam site [72–74]. The implementation of connectivity improvement through dam removal in the Douro catchment and especially in the Corgo sub-catchment is expected to make no exception.

Despite the potential benefits, dam removal can also trigger negative impacts. Exotic invasions in the lower stretches of rivers are often a significant pressure on native fish populations. The Douro River and tributaries make no exception. This means that dam removal could increase connectivity for exotic species as well as for native species. Previous studies have underlined how exotic invasions can substitute native fish species irrespectively of some hydrological conditions [75], and are a major driver of native fish distribution [76]. Furthermore, a combination of migration barriers and water abstraction have been found to have held exotic invasions in check, and even benefit some native species [77].

Some other authors highlight the fact that artificial lentic habitats created by dams can act as refuges for increasingly imperiled freshwater fishes [51], isolated from the invasive alien species. Besides these ecological issues, there are socio-political concerns about dam removal. In New England, for example, where over 14,000 dams fragment the region's rivers, dam removals are often highly contested. This is due, in part, to how the intertwined roles of history, identity, and aesthetics coalesce to create an attachment to places, and inspire the defense of dammed landscapes [50].

There are alternatives to dam removal, like the trap and transport option. This alternative refers to the trapping and transport of spawners to spawning sites upstream (or downstream in the case of juveniles). This option is currently used to migrate juvenile salmonids in large catchments [78], and has been tested with cyprinids in the Sabor River. The procedure allows identifying suitable spawning grounds and the optimal habitats for key target species [79], at the same time that it avoids the progression of invasive exotic species when the obstacle is removed. There is also a growing knowledge about deterrent systems that have been used in different management applications [80]. These deterrent systems comprise physical and nonphysical barriers developed to direct fishes into

the appropriate directions, or to guide them away from sources of mortality. Anyway, since all these techniques can only be implemented for a restricted number of structures, the present work is crucial to select them.

It is worth noting that there are many methods for estimating the permeability of an obstacle, which must be analysed according to the attributes of each barrier and the requirements and possibilities of the passage of each fish species in both directions [81]. The approach used in the present study was helpful in exposing a sub-sample of dams or weirs to visit where permeability needs to be carefully evaluated in the field, namely the dams and weirs characterized by a larger link improvement value, as they are promising candidates for dam removal. The cartographic assessment of permeability (Table 1) that allowed estimating the link improvement is supportive, because it would be rather costly and time consuming to characterize all of the 1193 obstacles detected by the satellite images. So, the next phase is to visit the structures ranked in the two or three highest classes of link improvement.

In the field characterization of dam or weir permeability, we will rely on the study entitled "Longitudinal connectivity diagnosis in the Douro River Basin", available at the website of Douro River Basin Authority (http://www.chduero.es/acciona5/metodologia/ic.pdf). This report comprises a connectivity index that ranges from 0 to 100 and implies the fields characterization of hydraulic conditions close to the obstacles. The procedure for obtaining values for this index requires that calculations are made separately for the various fish groups according to their specific swimming requirements. While adopting this index it has also ensured a common analysis of connectivity for the entire Douro catchment (integrating Portugal and Spain).

The ongoing field work is expected to enable more precise predictions of connectivity in all the watersheds based on the hydraulic characterization of critical dams or weirs that could be identified in the present work on the basis of a cartographic evaluation of connectivity. When the field refinement of dams and weirs permeability is complete, the permeability values depicted in Table 1 for each structure will be corrected, and CS26 will run with the new values, producing a more accurate view of connectivity in the entire Douro catchment. The final dPC connector and link improvement maps of the Douro River basin will be helpful for the River District authorities, not only as regards the obstacles that have to be considered for mitigating regularization, but also for the river segments that should be excluded from any additional damming.

In the present work we completed Step 3, which will be followed by steps 4–6 defined in the general workflow (Figure 2). These subsequent steps will deal with costs and benefits of alternative scenarios for decision making. A hierarchical decision-support framework to rank dams for removal will then be used based on guidelines and criteria similar to [82–84]. These works include prioritization scenarios that rank dams based on their suitability for removal using: Ecological, social (ecosystem services) and biodiversity criteria. Anyway, the present work provided a first insight into habitat patch connectivity in this important catchment, and constitutes the dorsal column of all planning.

**Supplementary Materials:** The following are available online at http://www.mdpi.com/2073-4441/11/4/693/s1, Barriers Database (for the Douro River Basin, Portugal).

**Author Contributions:** Conceptualization: R.M.V.C.; methodology: R.M.V.C. and A.P.; software: A.P. and J.P.M.; validation: F.A.L.P. and D.P.S.T.; formal analysis: R.M.V.C. and A.P.; investigation: R.M.V.C. and A.P.; resources: L.F.S.F.; data curation: J.J.B.J., M.P.M.M. and P.J.S.F.; writing—original draft preparation: R.M.V.C.; writing—review and editing: F.A.L.P., R.M.V.C. and D.P.S.T.; visualization: D.P.S.T. and J.P.M.; supervision: R.M.V.C. and F.A.L.P.; project administration: R.M.V.C. funding acquisition: R.M.V.C. and L.F.S.F.

**Funding:** This work was financed by the Project "Reviving Douro Basin", sponsored by MAVA, Foundation pour la Nature, and GEOTA—Grupo de Estudos de Ordenamento do Território e Ambiente. The authors integrated in the CITAB research centre were financed by National Funds of FCT–Portuguese Foundation for Science and Technology POCI-01-0145-FEDER-006958, under the project UID/AGR/04033/2019. The author integrated in the CQ-VR was funded by National Funds of FCT–Portuguese Foundation for Science and Technology POCI-01-0145-FEDER-006958, under the project UID/QUI/00616/2019.

**Conflicts of Interest:** The authors declare no conflict of interest. The funders had no role in the design of the study; in the collection, analyses, or interpretation of data; in the writing of the manuscript, or in the decision to publish the results.

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
