# Peer review of "Undamming the Douro River Catchment: A Stepwise Approach for Prioritizing Dam Removal"

_water, doi:10.3390/w11040693_

Round 1

Reviewer 1 Report

The topic of the article is certainly interesting. However the manuscript has several flaws that  the authors need to correct before this article can be published in Water.

First the language is not as polished it should be. There are some parts that are difficult to understand. I recommend the authors to carefully revise the text preferably with the aid of a native English speaker.

Second, the abstract needs to be improved. Consider to extend the abstract by mentioning more details about the methods and the results (e.g what is the probabilistic model you used?)

Then the introduction must be improved. Basically you only mention the benefits of the dam removal but your work is a methodological approach for prioritizing dam removal. You should add at least a paragraph about these methods.

Similarly methods can be improved, especially the part about the probabilistic model which is the most important part of your methodology

The presentation of the results is poor. There are some maps and tables and just few lines of text describing what you found

Apart from the aforementioned general issues below I list some more specific comments

Line 12: there is, not there are

Line 14-15: I recommend to rewrite this sentence. For instance you can simply say: In this study we assessed the inventory of the transversal barriers in the Douro catchment with the use of satellite imagery.

Lines 15-18: Not clear what you did. Please consider to elaborate. You can add some more information in the abstract. Besides it is quite small at it’s current form. You can increase the size of the abstract.

Why you use future tense in some occasions? You refer to what you will do in next steps of your research? I would suggest to remove that part. You should mention what you did and what this article is a about.

Line 24: Please specify river connectivity

Line 25: I think water demand is correct. Not water demanding

Line 26: Not sure that the word scale is the word you need here.

Line 27: agriculture demand? Maybe you mean agriculture intensification as a pressure?

Line 35: you start this paragraph by mentioning that there are impacts of dams on biota but the whole part is dedicated to the benefits of dam removal for rivers.

Line 36: diverse how?

Introduction must be improved. For instance you could mention the current status on the methods used for decision making on dam removal and then you can refer to the novelty and importance of your approach

Line 85: You need to explain what is this workflow. It is better to present this workflow schematically with a figure.

Line 132: typo error in transversal

Line 134: Please consider to elaborate the description of the probabilistic model. Maybe even add a separate subsection

Line 202: Do you mention this anywhere in the methods section? You should clarify how you defined the permeability conditions based on specific fish groups.

Author Response

Reviewer #1

General appreciation

The topic of the article is certainly interesting. However the manuscript has several flaws that the authors need to correct before this article can be published in Water.

Authors response: We very much appreciate the reviewer’s interest on the topic of our study as well as the effort put in the revision. We did our utmost to respond adequately to all comments and suggestions aiming your final approval.

Reviewer Comment #1

First the language is not as polished it should be. There are some parts that are difficult to understand. I recommend the authors to carefully revise the text preferably with the aid of a native English speaker.

Authors response: we substantially improved the language and style  in the revised version of the manuscript.

Reviewer Comment #2

Second, the abstract needs to be improved. Consider to extend the abstract by mentioning more details about the methods and the results (e.g what is the probabilistic model you used?)

Authors response: we substantially extended the abstract and mentioned the requested details.

Reviewer Comment #3

Then the introduction must be improved. Basically you only mention the benefits of the dam removal but your work is a methodological approach for prioritizing dam removal. You should add at least a paragraph about these methods.

Authors response: the reviewer is right. To improve the introduction section in the revised version we added one paragraph to identify the problem (damming) and them indicate the solution (dam removal and other alternatives not considered in the original version). Additionally, we added two paragraphs that describe the specificities of existing prioritization models, and indicate where we introduced improvements. We reproduce the sentences introduced in the revised version:

The problem:

“The construction of dams affects the biodiversity of riverine ecosystems in a number of manners. The hydrologic regime of streams is modified from lotic to lentic when stream water is retained in the reservoir, while this new environment frequently promotes the anomalous spreading of exotic species more adapted to the lentic conditions and concomitantly the fall of native species [1]. The cyclic streamflow variability of rivers is reduced by the presence of a new water body [2], which has been shown to increase water channel homogeneity and degrade aquatic fauna habitats [3]. The phytoplankton productivity and the thermal stratification of water tend to increase while dissolved oxygen decreases, resulting in a rapid increase of macronutrients in lake water [4,5] that temporarily increases the abundance of aquatic flora [6]. However, the progressive increase of nutrients in the reservoir deteriorates water quality, ultimately leading to eutrophication with negative consequences for the aquatic fauna [7–10].

Concurrently with the impact on stream flow regime and freshwater quality, the damming of rivers hinders the free circulation of migratory or resident species causing habitat fragmentation. The interruption of connectivity reduces the abundance of spawning sites compromising the survival capacity of juveniles [11]. This ability to survive can be further reduced downstream from the dam by hypolimnetic discharges and concomitant fluctuations in temperature and stream water composition [12,13].

The the alternative solutions:

The scenarios of generalized habitat fragmentation urge mitigation. A review on the ecological restoration of fluvial rivers has been recently published [17], which comprises techniques for rivers affected by engineering control. In general, the ecological restoration of rivers affected by construction of hydraulic facilities mainly includes two aspects: restoration of river’s geomorphological features and natural hydrological regime.  The restoration zones of river geomorphological features can be divided into two parts: riparian restoration and river channel restoration [18,19]. The natural hydrological regime can be restored through reconnecting abandoned channels to mainstreams, restoring the links between surface and groundwater flow to enhance vertical connectivity and communities associated with the hyporheic zone, and decommissioning of unsafe or obsolete dams, which can take the forms of full removal, partial removal of key components or abandonment [20]. New concepts on the spatial planning of sustainable water reservoirs [21,22] can also be used to reconcile water resources management and protection of biodiversity.

The models:

“The model by Kuby and co-workers [47], published in 2005, was focused on salmon migration in the Pacific Northwest region of the United States, and aimed to optimize ecological and economic loss objectives. The modeling of an ecological objective had the purpose to maximize the river and stream drainage area that has migratory access to the ocean. The economic loss objective was assumed to be purely illustrative in the model, and aimed to minimize two of the many economic services that are lost when a dam is removed: hydroelectric generating capacity, storage capacity. Among the specificities of Kuby’s model, one is noticeable: the function used to maximize the ecological function ensures that the benefits of removing a dam are not counted unless salmon would encounter no other dams between the ocean and the removed dam.

The more recent model of Hoenke, Kumar and Batt, published in 2014 [48], comprised a GIS tool for prioritizing dams for removal based on ecological criteria, social and ecological criteria taken altogether, and habitability of anadromous fish criteria. The results differed among the three model runs, but some dams ranked high in all runs the reason why were considered a top priority for removal. The tool includes a criterion for improving habitat connectivity based on three connectivity indicators: (a) functional upstream mileage, which is the number of miles upstream of a dam to the next barrier, (b) total upstream and downstream functional connectivity, which is the combined number of functional miles upstream and downstream of the dam, and (c) downstream barriers, which is the number of barriers downstream of a dam. A higher indicator rank is assigned for higher upstream mileage and total connectivity. For anadromous fish prioritizations, a higher rank is assigned for a lower number of downstream barriers.

The aforementioned and other similar models lack considerations about the permeability conditions across dam structures when the criterion for improvement of habitat connectivity is calculated. This would include, for example, information on the presence (or not) of passages for fish. The purpose of this study was therefore to improve the assessment of habitat connectivity through definition of a probabilistic function where permeability is taken into account.

Reviewer Comment #4

Similarly methods can be improved, especially the part about the probabilistic model which is the most important part of your methodology.

Authors response:  The methods section was substantially re-written and improved. Firstly, a general workflow (Figure 2a) was added to the revised version that clearly indicates the six model steps and discriminates the ones explored in the present study. Another workflow (Figure 2b) was also included to explain in detail the steps of compilation and processing of data towards calculation of connectivity indices.

The inclusion of workflows in the revised version complemented with the adjustment of text descriptions to these figures substantially improved the methods section style and comprehension. Please consult the revised Methods section to see the aforementioned changes.

Besides these changes, other text improvements were accomplished:

The exclusion of Main Water Course (Douro River) from the connectivity analysis is now explained in detail:

“This decision was considered inevitable but imposed some restrictions to the connectivity analysis. The issues and implications involved in this decision were: 1) the large Crestuma-Lever dam, installed in the Douro River close to the estuary, virtually impedes the migration of diadromous species such as salmon, savel or lamprey. The fish passages are ineffective and there is a sharp salinity contrast between the fresh river water upstream of the dam and the brackish water downstream. However, it is not considered the possibility to remove the dam because of its importance for hydropower generation. On the other hand, the use of Douro River is crucial for local socio-economic reasons, where tourism became the main driving force, and the dams contribute to that economy besides the other multiple purposes. It is therefore inevitable to keep the large infrastructures of Douro on site. Consequently, the current assessment on connectivity will not cover the benefits of dam removal for the Diadromous species. The rationale differs for Potamodrous species such as barbel, boga or trout, because the other Douro River dams are equipped with fish passages, which regardless the poor maintenance allow the movement of fish. In this case, the connectivity along the main water course is not compromised.”

The allocation of permeability values to obstacles is now explained in detail:

“For the present study, an important parameter to consider was the permeability of obstacles, classified in the 0–1 range. The classification of permeability was performed according to the criteria described in Table 1. The maximal value (one) represents full permeability and occurs when the fish can move freely along the water course (absence of barrier). Permeability decreases whenever the fish encounters a barrier. If the barrier is equipped with a fishway, the decrease was considered small and therefore the permeability value was reduced to 0.9. If the barrier was observed in the Google or Bing images but not an upstream reservoir, it was assumed that the obstacle allows stream flow and therefore the permeability value was reduced to 0.7. When the available information confirms the existence of a reservoir created by the barrier, but the reservoir could not be observed in the satellite images the permeability value was reduced to 0.5. Finally, for barriers with reservoirs clearly detected in the satellite images the permeability value was reduced to 0.3.

The raw data used in the calculation of PC, pPC connector and link improvement indices using CS26 is now clarified:

“The raw data for CS26 comprised the vector dataset with the drainage network characterized for stream length and the vector dataset with the barriers classified according to permeability (Figure 2b). These GIS datasets were used as input to the CS26 extension for ArcGIS, which produced the “nodes” and “links” text files as output. The “nodes” file contained the list of stream segments located between barriers and their lengths (a).  The “links” file contained the barrier permeability (p) retrieved from the barrier vector dataset. The two files were used as input to CS26, which calculated the connectivity indexes for the segments according and the link improvement values for the barriers, according to specific equations described below. Finally, these values were joined back to the original stream and barrier vector datasets to produce the thematic maps in ArcMap (e.g., Figure 3 below).

Reviewer Comment #5

The presentation of the results is poor. There are some maps and tables and just few lines of text describing what you found

Authors response: We thank this comment. In the revised version we extended the explanation about figures and tables.

As regards description of Figure 3 (former figure 2) we added the following sentences / paragraph:

“The majority of these obstacles (604) were installed in low order streams but quite a number of them (589) were placed in the main water courses of Douro catchment and principal sub-catchments (e.g., Tâmega, Sabor, Côa or Corgo). The construction materials were rock fill (912), concrete (280) or earth (1). The land use surrounding the dams was agriculture (905 cases), scrubland (3), forest (1) or mixed (284). The riparian gallery were mostly sparse (1181 cases) or inexistent (1), being continuous in just 11 cases. In general the dams or weirs were used for irrigation (1129 cases), while some dams corresponded to large (15) or small (51) hydroelectric plants.”

“The density of obstacles is not uniform across the studied area. In the Corgo (3.9 obstacles/100 km2) and Tâmega (3.22) basins the coverage with dams or weirs is much denser than the coverage in the Sabor (1.33) or Côa (1.19), but rainfall in the first two catchments (» 1500 mm/yr) is also much larger (twice) than in the ther catchments (» 750 mm/yr). Therefore, dam or weir installation seems to be related with rainfall availability rather than water scarcity.

As regards description of Figure 4 (former Figure 3) we added the following paragraph:

“This corresponded to the calculation of connectivity indices in 1497 nodes and 184 dams or weirs. The average node lengths ranged from 2404 m (Tâmega, 468 nodes), 2908 m (Sabor, 631 nodes), 4021 m (Côa, 305 nodes) and 2590 m (Corgo, 93 nodes). These results link longer nodes to the Côa basin and shorter to the Tâmega basin. The permeability indices ranged from p = 0.68±0.11 (Tamega, 83 barriers), p = 0.62±0.14 (Sabor, 48 barriers), p = 0.65±0.10 (Côa, 35 barriers), and p = 0.70±0.07 (Corgo, 18 barriers), meaning that they are relatively constant within the four basins.

Apart from the aforementioned general issues below I list some more specific comments

Line 12: there is, not there are

Answer: corrected

Line 14-15: I recommend to rewrite this sentence. For instance you can simply say: In this study we assessed the inventory of the transversal barriers in the Douro catchment with the use of satellite imagery.

Answer: Done

Lines 15-18: Not clear what you did. Please consider to elaborate. You can add some more information in the abstract. Besides it is quite small at it’s current form. You can increase the size of the abstract.

Answer: The abstract was substantially extended to provide a comprehensive view over what we did.

Why you use future tense in some occasions? You refer to what you will do in next steps of your research? I would suggest to remove that part. You should mention what you did and what this article is a about.

Answer: Well, this is an ongoing project with a very important step completed that merits publication. The identification of future work is not uncommon in papers and this study makes no exception. Besides, by including some information of the future work we also provide a general scope of the entire project, which is helpful for the readers

Line 24: Please specify river connectivity

Answer: done

Line 25: I think water demand is correct. Not water demanding

Answer: done

Line 26: Not sure that the word scale is the word you need here.

Answer: we rewrote the sentence as: “These changes are widespread in the Portuguese side of Douro River…”

Line 27: agriculture demand? Maybe you mean agriculture intensification as a pressure?

Answer: Yes. Intensification is the word. We replaced “demand” by “intensification”.

Line 35: you start this paragraph by mentioning that there are impacts of dams on biota but the whole part is dedicated to the benefits of dam removal for rivers.

Answer: The reviewer is right. We rewrote the sentence as follows: “Investigation on dam removal has become intense and spanned a diversity of topics in recent years.”

Line 36: diverse how?

Answer: in the new sentence (see previous comment), diverse was changed to a “diversity of topics”, described in the next sentences.

Introduction must be improved. For instance you could mention the current status on the methods used for decision making on dam removal and then you can refer to the novelty and importance of your approach.

Answer: We substantially improved the Introduction section as mentioned above.

Line 85: You need to explain what is this workflow. It is better to present this workflow schematically with a figure.

Answer: As mentioned above, in the revised version we added two workflows, one to portray the general steps of a dam removal prioritization model and another to illustrate the connectivity assessment model. Both models are now comprehensively described.

Line 132: typo error in transversal

Answer: corrected

Line 134: Please consider to elaborate the description of the probabilistic model. Maybe even add a separate subsection

Answer: the whole methods section was substantially re-written to elaborate the description of the probabilistic model. We believe the methods section is now much more comprehensive.

Line 202: Do you mention this anywhere in the methods section? You should clarify how you defined the permeability conditions based on specific fish groups.

Answer: We removed the sentence.

Reviewer 2 Report

Dear editor and authors,

I read and reviewed the MS by Cortes et al. entitled “Undamming the Douro River catchment: a stepwise approach for prioritizing dam removal”.

This rather short manuscript deals with a procedure to map and evaluate the River Douro habitat fragmentation. The authors have processed satellite imagery to pinpoint the location of some migration barriers in some sub-basins of this large Iberian river. Then proceeded to compute a connectivity index for each barrier, expressing how much it fragments the habitat. Ultimately this should theoretically help to prioritize the actual inspection of barriers in the field, and field removal.

I found several general shortcomings in this work, which I believe would need to be addressed prior to publication.

The aims of this work are clearly of local interest and in progress, but I believe they could be of wider relevance if an effort to frame and discuss them in a more general context is made. The authors seem not to worry about this aspect at all in their current introduction. Several rivers in Europe and worldwide face similar problems as those encountered in the Douro Basin, but the Introduction jumps straight to the Douro River already at the second sentence. Then expands on some background in the following paragraph. I suggest the Introduction is reorganized with more sense of structure and logic. First give a scope of the problem, and explain why it is a problem rather than going straight to dam removal. Explain that dam removal is one of diverse management options and then make the case for your work, trying to frame it in a wider perspective.

Discussion is also severely flawed. It starts by detailing future actions needed according to the plan, which could be better suited for a concluding remarks/recommendations section. It does not address at all any potential shortcomings of the methodology used. It addresses very shortly the outcomes of the present work and their ecological/management significance, I do understand it is preliminary work but an effort should be made to underline its significance if it is to be published in an international journal. Please consider revising your Discussion extensively and enriching it with more citations.

Your Methods have a few puzzling issues. First you excluded the main river course because dams are not under evaluation for removal there, but this conflicts a bit with your background and is not addressed in Discussion.. a high fragmentation in the main river means that sub-basin connectivity won’t help as much the overall basin connectivity. To me that sounds like quite a relevant thing to at least consider. Your methodology for identifying the migration barriers is poorly described. Copernicus maps seem not to have a sufficient resolution to identify correctly the type of barrier, you mention you drew “relevant information” from Google and Bing (do you mean aerial photography through Google?) but don’t say what or how and you also mention that you collected additional information but don’t specify how. As such I don’t believe your work can be easily reproduced or checked by anyone, you need to provide much more detail on this procedure, which could be one of the strong points of your work. Along the same lines, you state that you provided an “indicative” assessment of barrier permeability to calculate preliminary values. This methodology has evident shortcomings and biases, but it is a key element of your work and its consistency is not evaluated nor addressed anywhere in the manuscript. If your values are purely indicative as you state they cannot be useful to prioritize dam removal, but not even to guide your following field inspections. That would kinda undermine most of your results… I believe that they do have some significance, but how much that is remains debatable.

There are some very important missing topics in the Introduction/Discussion. The work is very theoretical so why not address alternatives and consequences of what you propose to do? As an example, one thing the authors would need to discuss is the fact that exotic invasions in the lower stretches of rivers are often a significant pressure on native fish populations.  I believe this is the case for many rivers in the Mediterranean area, and that the Douro River makes no exception. This means that dam removal could increase connectivity for exotic species as well as for native species. Previous studies have underlined how exotic invasions can substitute native fish species irrespectively of some hydrological conditions (Milardi et al. 2018, Long-term fish monitoring underlines a rising tide of temperature tolerant, rheophilic, benthivore and generalist exotics, irrespective of hydrological conditions. Journal of Limnology) and are a major driver of native fish distribution (Milardi et al. 2018, Run to the hills: exotic fish invasions and water quality degradation drive native fish to higher altitudes. Science of the Total Environment). Furthermore, a combination of migration barriers and water abstraction have been found to held exotic invasions in check and even benefit some native species (Gavioli et al. 2018, Exotic species, rather than low flow, negatively affect native fish in the Oglio River, Northern Italy. River Research and Applications). As your paper deals with dam removal, I believe it would be important for you to address the potential consequences of your proposed actions.

Given the length and scope of this review I have not provided comments on relatively smaller issues, which I am ready to address should a revised version be resubmitted.

Author Response

Dear editor and authors,

General appreciation

I read and reviewed the MS by Cortes et al. entitled “Undamming the Douro River catchment: a stepwise approach for prioritizing dam removal”.

This rather short manuscript deals with a procedure to map and evaluate the River Douro habitat fragmentation. The authors have processed satellite imagery to pinpoint the location of some migration barriers in some sub-basins of this large Iberian river. Then proceeded to compute a connectivity index for each barrier, expressing how much it fragments the habitat. Ultimately this should theoretically help to prioritize the actual inspection of barriers in the field, and field removal.

I found several general shortcomings in this work, which I believe would need to be addressed prior to publication.

Authors’ response

We very much appreciate the effort of this reviewer put in the analysis of our manuscript, and welcome all the comments and suggestions. They were all attended and carefully handled.

Comment #1

The aims of this work are clearly of local interest and in progress, but I believe they could be of wider relevance if an effort to frame and discuss them in a more general context is made. The authors seem not to worry about this aspect at all in their current introduction. Several rivers in Europe and worldwide face similar problems as those encountered in the Douro Basin, but the Introduction jumps straight to the Douro River already at the second sentence. Then expands on some background in the following paragraph. I suggest the Introduction is reorganized with more sense of structure and logic. First give a scope of the problem, and explain why it is a problem rather than going straight to dam removal. Explain that dam removal is one of diverse management options and then make the case for your work, trying to frame it in a wider perspective.

Authors’ response

Many thanks for this pertinent comment. The Introduction section was reorganized according to this suggestion.

To give a scope of the problem, and explain why it is a problem we introduced the following paragraphs and associated references:

“The construction of dams affects the biodiversity of riverine ecosystems in a number of manners. The hydrologic regime of streams is modified from lotic to lentic when stream water is retained in the reservoir, while this new environment frequently promotes the anomalous spreading of exotic species more adapted to the lentic conditions and concomitantly the fall of native species [1]. The cyclic streamflow variability of rivers is reduced by the presence of a new water body [2], which has been shown to increase water channel homogeneity and degrade aquatic fauna habitats [3]. The phytoplankton productivity and the thermal stratification of water tend to increase while dissolved oxygen decreases, resulting in a rapid increase of macronutrients in lake water [4,5] that temporarily increases the abundance of aquatic flora [6]. However, the progressive increase of nutrients in the reservoir deteriorates water quality, ultimately leading to eutrophication with negative consequences for the aquatic fauna [7–10].

Concurrently with the impact on stream flow regime and freshwater quality, the damming of rivers hinders the free circulation of migratory species causing habitat fragmentation. The interruption of connectivity reduces the abundance of spawning sites compromising the survival capacity of juveniles [11]. This ability to survive can be further reduced downstream from the dam by hypolimnetic discharges and concomitant fluctuations in temperature and stream water composition [12,13].

To explain that dam removal is one of diverse management options we introduced the following paragraph and associated references:

“The scenarios of generalized habitat fragmentation urge mitigation. A review on the ecological restoration of fluvial rivers has been recently published [17], which comprises techniques for rivers affected by engineering control. In general, the ecological restoration of rivers affected by construction of hydraulic facilities mainly includes two aspects: restoration of river’s geomorphological features and natural hydrological regime.  The restoration zones of river geomorphological features can be divided into two parts: riparian restoration and river channel restoration. The natural hydrological regime can be restored through reconnecting abandoned channels to mainstreams, restoring the links between surface and groundwater flow to enhance vertical connectivity and communities associated with the hyporheic zone, and decommissioning of unsafe or obsolete dams, which can take the forms of full removal, partial removal of key components or abandonment [18]. New concepts on the spatial planning of sustainable water reservoirs [19,20] can also be used to reconcile water resources management and protection of biodiversity. It is worth to note, however, that the implementation of connectivity carries out social and political complex decisions, which can also be technically challenging because of the unique dendritic spatial structure of river systems [21–23].”

Comment #2

Discussion is also severely flawed. It starts by detailing future actions needed according to the plan, which could be better suited for a concluding remarks/recommendations section. It does not address at all any potential shortcomings of the methodology used. It addresses very shortly the outcomes of the present work and their ecological/management significance, I do understand it is preliminary work but an effort should be made to underline its significance if it is to be published in an international journal. Please consider revising your Discussion extensively and enriching it with more citations.

Authors’ response

We thank the reviewer’s comment. The discussion was extended and enriched with various other references. It also explicitly addressed the potential shortcomings of the method, namely as regards the exotic problem mentioned by the reviewer in a comment below.

Comment #3.1

Your Methods have a few puzzling issues. First you excluded the main river course because dams are not under evaluation for removal there, but this conflicts a bit with your background and is not addressed in Discussion. A high fragmentation in the main river means that sub-basin connectivity won’t help as much the overall basin connectivity. To me that sounds like quite a relevant thing to at least consider.

Authors’ response

The reviewer is right. In the revised version we expanded the justification to exclude the main river course and provided an indication on the implications of that decision for connectivity analysis. We reproduce below the revised text:

“The main water course (the River Douro) was excluded from the analysis. This decision was considered inevitable but imposed some restrictions to the connectivity analysis. The issues and implications involved in this decision were: 1) the large Crestuma-Lever dam, installed in the Douro River close to the estuary, virtually impedes the migration of Diadromous species such as salmon, savel or lamprey. The fish passages are ineffective and there is a sharp salinity contrast between the fresh river water upstream of the dam and the brackish water downstream. However, it is not considered the possibility to remove the dam because of its importance for hydropower generation. On the other hand, the use of Douro River is crucial for local socio-economic reasons, where tourism became the main driving force, and the dams contribute to that economy besides the other multiple purposes. It is therefore inevitable to keep the large infrastructures of Douro on site. Consequently, the current assessment on connectivity will not cover the benefits of dam removal for the Diadromous species. The rationale differs for Potamodrous species such as barbel or boga, because the other Douro River dams are equipped with fish passages, which regardless the poor maintenance allow the movement of fish. In this case, the connectivity along the main water course is not compromised.”

Comment #3.2

Your methodology for identifying the migration barriers is poorly described. Copernicus maps seem not to have a sufficient resolution to identify correctly the type of barrier, you mention you drew “relevant information” from Google and Bing (do you mean aerial photography through Google?) but don’t say what or how and you also mention that you collected additional information but don’t specify how. As such I don’t believe your work can be easily reproduced or checked by anyone, you need to provide much more detail on this procedure, which could be one of the strong points of your work.

Authors’ response

The author is right. In the revised version we substantially improved the methodological section. Firstly we introduced a workflow (Figure 2b), to clarify the operational procedure behind the connectivity assessment, from data compilation until calculation of the indices. Then, we adjusted the text according to the diagram. We believe the revised version is leaner and stronger as regards the presentation of methodology. Please see the yellow colored paragraphs throughout the revised section 2.2, which highlight the changes made. See also the new figures (2a and 2b) that clarify the whole context and procedures.

Comment #3.3

Along the same lines, you state that you provided an “indicative” assessment of barrier permeability to calculate preliminary values. This methodology has evident shortcomings and biases, but it is a key element of your work and its consistency is not evaluated nor addressed anywhere in the manuscript. If your values are purely indicative as you state they cannot be useful to prioritize dam removal, but not even to guide your following field inspections. That would kinda undermine most of your results… I believe that they do have some significance, but how much that is remains debatable.

Authors’ response

The term indicative was wrongly used.  In the revised version we produced the following improvements. Firstly, we described better the rationale for setting up the p values:

“For the present study, an important parameter to consider was the permeability of obstacles, classified in the 0–1 range. The classification of permeability was performed according to the criteria described in Table 1. The maximal value (one) represents full permeability and occurs when the fish can move freely along the water course (absence of barrier). Permeability decreases whenever the fish encounters a barrier. If the barrier is equipped with a fishway, the decrease was considered small and therefore the permeability value was reduced to 0.9. If the barrier was observed in the Google or Bing images but not an upstream reservoir, it was assumed that the obstacle allows stream flow and therefore the permeability value was reduced to 0.7. When the available information confirms the existence of a reservoir created by the barrier, but the reservoir could not be observed in the satellite images the permeability value was reduced to 0.5. Finally, for barriers with reservoirs clearly detected in the satellite images the permeability value was reduced to 0.3.”

Secondly, re rephrased the sentence where the term “indicative” was used. We also explained why we consider adequate the use of Table 1 values to measure permeability:

A list of patch probabilities (pij) representing a diversity of permeability conditions for fish were depicted in Table 1. These values are valuable indicators of permeability and are adequate for the evaluation of connectivity in the regional scale for planning purposes (current status). The evaluation of connectivity in the local scale and for decision making purposes requires the confirmation of pij based on field work, which is in progress.

Comment #4

There are some very important missing topics in the Introduction/Discussion. The work is very theoretical so why not address alternatives and consequences of what you propose to do? As an example, one thing the authors would need to discuss is the fact that exotic invasions in the lower stretches of rivers are often a significant pressure on native fish populations.  I believe this is the case for many rivers in the Mediterranean area, and that the Douro River makes no exception. This means that dam removal could increase connectivity for exotic species as well as for native species. Previous studies have underlined how exotic invasions can substitute native fish species irrespectively of some hydrological conditions (Milardi et al. 2018, Long-term fish monitoring underlines a rising tide of temperature tolerant, rheophilic, benthivore and generalist exotics, irrespective of hydrological conditions. Journal of Limnology) and are a major driver of native fish distribution (Milardi et al. 2018, Run to the hills: exotic fish invasions and water quality degradation drive native fish to higher altitudes. Science of the Total Environment). Furthermore, a combination of migration barriers and water abstraction have been found to held exotic invasions in check and even benefit some native species (Gavioli et al. 2018, Exotic species, rather than low flow, negatively affect native fish in the Oglio River, Northern Italy. River Research and Applications). As your paper deals with dam removal, I believe it would be important for you to address the potential consequences of your proposed actions.

Authors’ response

We very much thank this comment. We improved the introduction by mentioning alternative solutions to dam removal. We improved the discussion by expanding the comments on our results and enriching the section with more references (comparison with other studies). We also explicitly addressed the shortcomings of the method, namely the exotic issue, and included the aforementioned references in the revised version.

Comment #5

Given the length and scope of this review I have not provided comments on relatively smaller issues, which I am ready to address should a revised version be resubmitted.

Authors’ response

The smaller comments will be welcome.

Reviewer 3 Report

This is a good analysis; and it carefully describes part of a workflow to assess impacts. This is an important descriptive work that sets a practical and rapid method for assessment using fish permeability. The overall merit of the work is high since it is a conservation-relevant method and shows the potential for application. 

I would definately recommend a conceptual diagram (flow-chart_ within the workflow process section. This can show the step-wise procedure and highlight that part that is described in this paper.

There are many grammatical and other mistakes in the English language but this paper reads well and is straightforward. However I am particulalry bothered with the use of some terms that are not used widely in this particular field. For example: "transversal obstacles" is used in geometry but is probably not used in longitudinal studies of rivers or other lotic systems; please replace this with an appropriet phrase. Also, "fish transponibility" is a word used in the latin languages, but should not be used as such in English, please replace with the correct widely used term for this issue. 

Author Response

General appreciation

This is a good analysis; and it carefully describes part of a workflow to assess impacts. This is an important descriptive work that sets a practical and rapid method for assessment using fish permeability. The overall merit of the work is high since it is a conservation-relevant method and shows the potential for application. 

Authors’ response

We thank the reviewer for the positive appreciation on our work. We put the strongest effort in the revision to attend all comments and suggestions.

Comment #1

I would definately recommend a conceptual diagram (flow-chart_ within the workflow process section. This can show the step-wise procedure and highlight that part that is described in this paper.

Authors’ response

In the revised version we added two new figures. Figure 2a is a conceptual workflow describing the general steps involved in the dam removal prioritization model. Figure 2b is an operational workflow describing the steps used to calculate connectivity, from data compilation to index calculation. The methodological section as then improved and re-written according to these two workflows, becoming much clearer.

Comment #2

There are many grammatical and other mistakes in the English language but this paper reads well and is straightforward. However I am particulalry bothered with the use of some terms that are not used widely in this particular field. For example: "transversal obstacles" is used in geometry but is probably not used in longitudinal studies of rivers or other lotic systems; please replace this with an appropriet phrase. Also, "fish transponibility" is a word used in the latin languages, but should not be used as such in English, please replace with the correct widely used term for this issue.

Authors’ response

We replaced the terms mentioned by the reviewer with more appropriate terms.

Round 2

Reviewer 1 Report

The revised version is clearly improved. The authors have addressed all my comments and answered my questions regarding their work. Most important, the revised manuscript has been significantly improved, is easier to read than the previous version and the overall quality is higher.

Based on the above i recommend to accept the manuscript for publication at it's current form

Author Response

We very much appreciate the nice comments of the reviewer about our latest version of the manuscript. We did our best to improve the paper according to the reviewers comments and thank their recognition.

Reviewer 2 Report

I greatly appreciated the efforts by the authors to edit their manuscript according to the comments received. I think most of the general issues I had raised in my previous review were adequately addressed in their revision and that the manuscript seems substantially improved. In particular, I really liked the new figures and believe that the text now is richer and more balanced.

In this review, I’ll provide only few comments on relatively minor details, which I suggest the authors to take into account but otherwise do not preclude publication.

I think you should mention the scientific names of species upon first mention of their common name e.g. line 203. International readers won’t know what you mean with barbel (only the Barbus genus has more than 20 species..) or any other fish common name (e.g. salmon, trout etc. are very common fish names worldwide, consider at least writing their real common names, i.e. Atlantic salmon or brown trout).

I’m not myself a native speaker, but I did detect a number of language issues in the paper. I would suggest that the authors ask one native speaking colleague to have one last look through the paper and correct their phrase construction. I don’t think there’s any need for professional copyediting, but this would surely increase the readability of your work.

Author Response

Reviewer comment

I greatly appreciated the efforts by the authors to edit their manuscript according to the comments received. I think most of the general issues I had raised in my previous review were adequately addressed in their revision and that the manuscript seems substantially improved. In particular, I really liked the new figures and believe that the text now is richer and more balanced.

Authors answer

We very much appreciate the nice comments made by the reviewer about our revised manuscript.

Reviewer comment

I think you should mention the scientific names of species upon first mention of their common name e.g. line 203. International readers won’t know what you mean with barbel (only the Barbus genus has more than 20 species..) or any other fish common name (e.g. salmon, trout etc. are very common fish names worldwide, consider at least writing their real common names, i.e. Atlantic salmon or brown trout).

Authors answer

the real common names were included in the revised verion (Atlantic Salmon) and European river or brook lampreys.

Reviewer comment

Im not myself a native speaker, but I did detect a number of language issues in the paper. I would suggest that the authors ask one native speaking colleague to have one last look through the paper and correct their phrase construction. I don’t think there’s any need for professional copyediting, but this would surely increase the readability of your work.

Authors answer

The revised version was thoroughly revisited to detect grammar tenses or incorrect spelling. We believe everything was corrected.